# Project COALESCE—An Example of Academic Institutions as Conveners of Community-Clinic Partnerships to Improve Cancer Screening Access

**DOI:** 10.3390/ijerph19020957

**Published:** 2022-01-15

**Authors:** Katherine Y. Tossas, Savannah Reitzel, Katelyn Schifano, Charlotte Garrett, Kathy Hurt, Michelle Rosado, Robert A. Winn, Maria D. Thomson

**Affiliations:** 1Department of Health Behavior and Policy, School of Medicine, Virginia Commonwealth University, Richmond, VA 23298, USA; reitzelsh@vcu.edu (S.R.); maria.thomson@vcuhealth.org (M.D.T.); 2Division of Epidemiology, Department of Family Medicine, School of Medicine, Virginia Commonwealth University, Richmond, VA 23298, USA; 3Massey Cancer Center, Virginia Commonwealth University, Richmond, VA 23298, USA; katelyn.schifano@vcuhealth.org (K.S.); cllitzenberg@vcu.edu (C.G.); khurt@vcu.edu (K.H.); michelle.rosado@vcuhealth.org (M.R.); Robert.Winn@vcuhealth.org (R.A.W.)

**Keywords:** quality improvement collaborative, didactic partnerships, early detection of cancer, colorectal neoplasm, cervical neoplasm, implementation science, community health centers, community organizations, academic-community partnership, community outreach and engagement

## Abstract

In Virginia, 56% of colorectal cancers (CRC) are diagnosed late, making it one of three enduring CRC mortality hotspots in the US. Cervical cancer (CCa) exhibits a similar pattern, with 48% late-stage diagnosis. Mortality for these cancers is worse for non-Latinx/e(nL)-Black people relative to nL-White people in Virginia, but preventable with equitable screening access and timely diagnostic follow-up. However, structural barriers, such as fractured referral systems and extended time between medical visits, remain. Because Federally Qualified Health Centers (FQHCs) care for a large proportion of racial and ethnic minorities, and underserved communities, regardless of ability to pay, they are ideal partners to tackle structural barriers to cancer screenings. We piloted a quality improvement initiative at five FQHCs in southcentral Virginia to identify and address structural, race-related barriers to CRC, as well as CCa screening and diagnostic follow-up using evidence-based approaches. Uniquely, FQHCs were paired with local community organizations in a didactic partnership, to elevate the community’s voice while together, increase support, acceptance, uptake, and intervention sustainability. We report on project development, and share preliminary data within the context of project goals, namely, to increase cancer screenings by 5–10%, improve knowledge and diagnostic follow-up processes, and build longitudinal partnerships.

## 1. Introduction

With 56% of colorectal cancers (CRC) diagnosed at later stages, it is no surprise that Virginia (VA) remains one of three enduring CRC mortality hotspots in the United States [1]. Black Virginians are disproportionately affected by CRC, recording the highest CRC incidence and mortality compared to other races and ethnicities [2]. Though less common, cervical cancer (CCa) in VA is not much different: 48% of CCa cases statewide are diagnosed at later stages, and CCa incidence and mortality in VA is greater in those who identify as Black and/or Latinx/e compared to other races [3]. Yet, prevention and early detection of these cancers is possible through screenings that detect and/or remove pre-cancerous lesions (Papanicolaou (Pap) for CCa, and stool-based (i.e., FIT or FOBT) or visual (structural) exams (i.e., colonoscopy, flexible sigmoidoscopy) for CRC). Therefore, these elevated rates of late-stage diagnoses and death are unacceptable. 

Much attention has been given toward addressing individual and interpersonal level barriers to cancer screening, such as increasing opportunities for provider recommendation, and improving patient knowledge to address awareness, receptivity, and fear [4,5]. However, tackling existing racial and ethnic disparities in CRC and CCa requires acknowledgment of and intervention on structural barriers that eclipse an individuals’ range of opportunities to access routine screening and timely diagnostic follow-up. These structural barriers are non-economic obstacles or burdens that make it difficult for individuals to access the care they need [6]. Acknowledgement of the extent and injustice of existing structural barriers has intensified in parallel with growing socio-political and racial unrest in the US, and the disproportionate morbidity and mortality experienced by minority communities during the COVID-19 pandemic. Community-engaged approaches can aid in the identification of key structural barriers driving differential access to screening and timely diagnostic follow-up. 

The principles of community engagement are based on a core belief that academics work alongside communities to ensure a shared agenda and a respectful understanding of the community context, while cultivating strong relationships with key partners, and leveraging available resources [7]. Public health has an extensive history of using community engaged approaches to guide co-development and execution of programs for improved effectiveness [8]. However, academic–community partnerships are almost always exclusively instigated by the academic partners, often in response to grant funding opportunities or investigator-initiated research interests. Seldom is the community able to join into a truly equal partnership, contribute their community intelligence, and potentially provide a deeper understanding of their shared community. Such partnerships can ultimately lead to higher trust, uptake, innovation, and sustainability of initiatives, and ensure that chosen initiatives truly represent the communities’ need and priorities. 

In this manuscript, we describe the process of development for Project *COALESCE*—Clinics and Communities Tackling Racial Disparities Structural in Cancer Screening. The purpose of COALESCE was to facilitate the development and execution of a quality improvement initiative (QI) within Federally Qualified Health Centers (FQHCs), to identify and address structural, race-related barriers to CRC and CCa screening and diagnostic follow-up. Uniquely, Project COALESCE would facilitate a partnership between FQHCs and local community organizations to work together (in a didactic, equal partnership) to identify the targets, design a plan of action, and disseminate the QI initiative. FQHCs are community-based clinics that receive funds from the Health Resources and Services Administration (HRSA) to provide primary care regardless of a patient’s ability to pay. The goals of COALESCE were to: (1) increase CRC and CCa screenings by 5–10% at each FQHC; (2) improve diagnostic follow-up processes; (3) increase provider and community cancer screening knowledge; (4) build longitudinal partnerships that support cancer health equity initiatives beyond the term of the project. The following is a description of our experiences developing and facilitating this adapted model of academic–community partnerships. 

## 2. Materials and Methods

### 2.1. Project Overview

Project COALESCE was designed as a small, feasibility pilot to stimulate a mutually beneficial partnership between an FQHC and a non-clinical community partner organization to identity a structural barrier and an evidence-based intervention (EBI) to implement towards improving CRC and CCa screening compliance. Uniquely, in this design, the role of the academic partner was strictly facilitation of the partnership, that is, convening regular meetings, the provision of training, and implementation support. Equalizing QI knowledge, and allowing the choice of EBI by each dyad were critical design elements, as clinics and community organizations best understand how and what would be best adapted and implemented within their communities. Working jointly (as a dyad), the FQHC and their community partner assessed their own practices, processes, resources, and perceived barriers pertaining to these cancer screenings, and selected an EBI to jointly implement towards increasing screening rates. The intention was that the dyadic approach would leverage exploration of critical barriers from the clinic and community perspectives, and begin to build robust partnerships directed by patient and community need rather than the priorities of the FQHC or the academic organization. 

### 2.2. Identification and Recruitment of FQHCs

As this was a feasibility pilot, we used 2018 HRSA [9] data to identify five FQHCs in Virginia with CRC and CCa screening rates well below the state’s average (70% and 84%, respectively), and the Healthy People 2030 goals [10] for these cancers (74% and 84%, respectively). We ensured these FQHCs minimally represented or surpassed the racial and ethnic (19% Black, 10% Latinx), and geographic (25% rural) diversity of the state. Recruitment consisted of a phone call to the targeted FQHC leadership, and a direct conversation with either the Chief Executive Officer, Chief Medical Officer, or equivalent, to explain the purpose and goal of the project. Importantly, none of the FQHCs recruited had formal contractual agreements with Virginia Commonwealth University (VCU) Health or VCU Massey Cancer Center at the time of recruitment. All FQHCs approached agreed to participate, provided letters of support for the initial grant submission, and signed subcontracts with VCU for the purposes of grant fund disbursement.

### 2.3. Identification and Recruitment of Community Partner Organizations

The partner non-clinical community organizations were either identified by the FQHC based on their existing relationship (for two of the FQHCs), or recommended by the project PI based on existing relationships with the Virginia Commonwealth University Massey Cancer Center (MCC), Office of Community Outreach and Engagement (for three FQHCs). Selected community organizations had aligned interests to improve the health of their communities, including access to cancer screenings and follow-up cancer care. They also served in the same patient service area and communities served by the partner FQHC. They were to join as equal accountability and didactic partners to the FQHC, assisting in an honest assessment of needs, development of plans, and implementation of adapted EBIs based on their collective findings. The FQHCs and community organizations were acknowledged as equal partners not only during their recruitment, contractual process, and training, but importantly, their equal partnership was also acknowledged by allocating an equal amount of grant dollars to compensate for their participation. The joint responsibility was also acknowledged by adjoining their deliverables such that the success of the QI initiative was dependent on their equal participation, measured by templated reports (co-developed during the recruitment process) that included joint evaluations. The submission of this (biennial) report would trigger grant disbursements to participating entities.

### 2.4. Identification of Targets for Program Evaluation Metrics

The primary target for the program was to increase screening rates by 5–10% in 24 months, which was agreed upon by each FQHC as a reasonable goal. To estimate the approximate number of women eligible for CCa screening (denominator), the reported total patient population was multiplied by the proportion of adults aged 18–64, assuming a 50/50 split of the population by gender. We then used the HRSA-reported CCa screening average for the five participating FQHC, estimated at 49.6%. Based on this, we estimated having to screen 3000 age-eligible women to increase this average CCa screening rate across the five clinics from 49.6% to 60% over a 24-month period, consistent with the length of the project. Using the same estimation and the proportion of adults 65+ reported in HRSA, we estimate having to screen around 1300 individuals to increase the average CRC screening rate across the five clinics from 38.2% to 50% over the project period. All participating FQHCs agreed to provide up to date (CY2020) baseline data that would allow us to estimate their latest, actual CCa and CRC screening rates overall, and by race/ethnicity. They also provided data on the number of clinical and non-clinical staff, and information on the availability of various services relevant to cancer screening access (i.e., extended clinic hours, copay sharing, walk-in CCa screenings, EMR, auto-reminders, etc.) as part of the initial needs assessment. Finally, they also agreed to submit the same data at 12 and 24 months, inclusive of the number of abnormal screenings found during the project, to estimate potential cancers averted. The community organizations provided a total number of members, overall, and by race and ethnicity, as well as broad information regarding their focus and services.

### 2.5. Process Design and Methods

We scheduled monthly didactic-style one-hour meetings to train the FQHCs and partner community organizations (referred to as dyads) on various quality improvement (QI) tools, and to exchange information on processes and progress towards stated common goals. Upon receiving training, dyads used the QI tools to jointly identify and address barriers to CRC and CCa screening as follows: first, dyads conducted an environmental scan (ES) using an adaptation of the CDC-recommended University of Kentucky’s seven-step process [11] to broadly assess activities related to CRC and CCa screening, not only in the FQHC, but also those executed by the partner community organizations where applicable. The ES included a strengths, weaknesses, opportunities, and threats (SWOT) analysis [12], and findings were summarized across all dyads. Next, dyads conducted a guided root cause analysis using the Fishbone (Ishikawa) diagram [13] to jointly identify and prioritize barriers to screening and diagnostic follow-up access, especially those that may be differentially impacting racial/ethnic minority populations. Finally, dyads selected a solvable problem or process (a “bone” from the fishbone) to jointly address using Plan, Do, Study, Act (PDSA) cycles [14]. 

We additionally conducted semi-structured qualitative interviews with members of the leadership, clinical and non-clinical staff of the FQHC, and members of the community organization to describe how each dyad developed and worked as a team. We also deployed two surveys: the first survey was for providers only, and consisted of a subset of questions from the National Survey of Primary Care Physicians’ Cancer Screening Recommendations and Practices [15]. The purpose of this survey was to assess cancer screening knowledge among clinical providers at onset, and again at the conclusion of the project, to measure change. The second survey was a brief version of the Index of Race-Related Stress (IRRS) [16]. This tool is intended to capture experiences of stress associated with common, day-to-day race-related experiences for nL-Black Americans, under the common knowledge that racism and race-based structures influence how Blacks and other marginalized communities perceive and engage with daily encounters. Though the IRRS was validated for use in nL-Black Americans, given the common knowledge that Latinx/e communities also experience race-based stress and discrimination, this survey was also translated and distributed in Spanish to reduce language barriers, and achieve a more comprehensive perspective, particularly for one of the community organizations with a majority Spanish-speaking staff (partnered with an FQHC with 33% Latinx/e patient population). Within the context of this initiative, given the proposed didactic partnership between FQHCs and grassroots community organizations to jointly conduct QI processes, we deployed this tool to all participants at baseline as a proxy for power differentials, and to potentially assess whether differences in self-reported experiences of race-related stress would correlate with successful implementation of this QI initiative (measured as change in screening rate, provider knowledge, and perceived trust [17] from their reports). 

Each monthly meeting featured a QI subject matter expert (SME) to discuss each of the above-mentioned tools. Additionally, we brought SMEs to discuss available CRC and CCa screening EBIs, and explain their science, successes, and opportunities across various communities. We proposed using the Interactive Systems Framework for dissemination and implementation to identify the “how to” gaps that influence what works in the field for the benefit of the public [14]. Attendance to monthly meetings is required and open to the FQHC, the community organization, and any community member at large, all who collectively provide their input into their perceived barriers, facilitators, and which EBI might be most appropriate for their specific target community. In addition to the monthly convenings, dyads meet on their own, bi-weekly, for a 30-min informal discussion/conversation about due dates, discussion on materials learned during the monthly meetings, progress, and next steps, facilitated by their appointed VCU/MCC project liaison. 

### 2.6. Monitor and Measure Project Milestones

We developed a minimally burdensome, maximally informative report template, with the participant’s input. This template was informed by validated tools, such as the TeamSTEPPS patient safety approach used in healthcare to improve communication and teamwork skills among healthcare professionals [18], and a health partnership trust scale [17]. Each dyad submits these reports biennially to (1) track progress of each participating dyad, and troubleshoot concerns or problems by providing the necessary supports, to ensure project success; (2) serve as an institutional trigger for incentive fund disbursement; (3) allow dyads to qualitatively reflect on their progress, and (power) dynamics, with questions such as “What went well?”, “What could go better?”, “What should improve?”, “What was learned from the discussions and accomplished deliverables?”, “What are some obstacles and facilitators to changes?”, “Did the partner feel their ideas/opinions were heard”, “Was there clear communication?”, and “Do participants sense their work will make a difference?”. 

### 2.7. Data Analysis

Quantitative data will be summarized and analyzed descriptively using t-tests for continuous variables, chi-squared tests for proportional and categorical variables, all at alpha 0.05 as the statistical significance threshold, and using StataIC 13 Stata Statistical Software: Release **13** (StataCorp LP, College Station, TX, USA). Qualitative data (such as SWOT analysis, qualitative interviews, and narrative data from reports) will be analyzed using a grounded theory approach to identify common, emerging themes. As data collection is ongoing, herein we provide the initial, univariate analysis of baseline data for year one of the project. 

## 3. Results

FQHC Baseline Data—HRSA, 2019 [9], (Table 1): Participating FQHC networks reported serving 87,419 patients (representing 26% VA’s FQHC patient population). Compared to non-participating FQHCs (*n* = 21), the five participating FQHCs served a higher proportion of patients that were younger than 65 (74% vs. 60%, *p* = 0.01), uninsured (43% vs. 28%, *p* = 0.04), and had double the proportion of nL-Black patients than non-participating FQHCs (46% vs. 23%, *p* = 0.05). Though the HRSA-reported CRC and CCa screening rates for the five participating FQHCs (38% and 50%) did not significantly differ from the same screening rates for non-participating FQHCs (40% and 47%), rates for all FQHCs were well below the state (70% and 91%), and the Healthy People 2030 screening goals (74% and 84%) for these cancers, respectively [10]. 

### 3.1. FQHC Baseline Data—Updated for Participating Clinics

FQHCs with multiple clinics chose one or several clinics to participate in this initiative, and submitted updated (CY2020) respective baseline data specific to those clinics (Table 2). These participating clinics reported serving 47,707 patients in total, which were a majority female (58%), non-White (47% nL-Black, 12% Latinx/e), and publicly insured (53% Medicaid/Medicare). They reported employing 170 non-clinical staff, 103 nurses, and 68 providers, a majority (82% of providers) of which performed on-site Pap screening for CCa, and all provided stool cards (predominantly Fecal Immunochemical Test (FIT) cards) for CRC. Of two clinics who reported the racial/ethnic composition of their providers, the clinics with the second and third largest proportion of nL-Black patient populations (62% and 44% nL-Blacks) reported that 0% and 13% of their providers, respectively, identified as nL-Blacks. The baseline CRC screening rate for participating clinics was 41%, and ranged from 20% to 71%. Males were less likely to screen for CRC compared to females (39% vs. 43%, respectively, *p* = 0.002). Though not overall statistically different by race/ethnicity, CRC screening rates were lowest for Latinx/e (37%), Asians (24%), and “Other” racial groups (22%) compared to nL-Whites (42%) and nL-Blacks (43%). For CCa, the baseline screening rate was 47%, ranging from 33% to 57%, with statistically relevant differences by race and ethnicity (*p* = 0.003), and, in this case, reporting highest CCa screening compliance for Latinx/e (54%), followed by nL-Whites (49%), nL-Blacks (45%), Other (41%), and the lowest for Asians (37%) (Table 2).

### 3.2. Community Organization Baseline Data

The five partner community organizations reported a total of 82 members, predominantly representing their leadership (i.e., board members) and/or staff (i.e., lay-health educators, community health workers, and other administrative staff, depending on the size of the community organization). These members were a majority female (89%), non-White (48% nL-Black, 39% Latinx/e, 5% Other), and over 50 years of age (50%). Their services ranged from information dissemination/education, to lay navigation, financial or transportation assistance, and language translation. Their identified constituencies ranged from their local county/city to adjacent (collar) cities and counties. All identified having a health equity focus, and a presence in predominantly Black communities.

### 3.3. Baseline Results from QI Tools: Environmental Scan

In SWOT analyses, there were three common themes identified by the dyads as “Strengths”. These focused on the availability of: (1) clinical support tools, such as Electronic Medical Records (EMR), with the ability to send messages, alerts, and reminders; (2) patient access factors, such as ability to offer a sliding scale, multiple clinic locations, evening and/or extended hours, and bilingual staff; (3) a focus on compassionate patient education provided by nurses and doctors, as well as fostering small town-type relationships, including relationships with lay health promoters. For “weaknesses”, the two common themes were: (1) patients’ lack of screening awareness, lack of knowledge about the need or timing of cancer screenings, and deep-seeded stigmas associated with cancer screenings; (2) internally, they identified information gaps impeding clinic flow, such as information missing from screenings done off site, a lack of resources to cover staff time for follow-up (impacting stool card return), and underutilization of EMR flagging of eligible patients, potentially due to inefficient set-up. For “opportunities”, there were three emerging common themes: (1) a common desire to improve communication with specialists external to their clinics; (2) numerous site-specific ideas and approaches for improving patient education to increase understanding and dispel fear; (3) opportunities to improve in-house automated systems to highlight/focus on prevention/early detection opportunities. For “threats”, though there were far fewer discussions, an identified common theme was patients’ fear of the procedure and the diagnosis, a concern that aligned with the “weakness” discussion. Fishbone: Three of the five dyads identified deficiencies in their own outreach and education approaches as a critical potential root cause of their low cancer screening rates. The other two dyads identified deficiencies in their internal follow-up processes as a potential root cause of their low cancer screening rates. PDSA: Respectively, three dyads developed a plan to address the identified outreach and education problem in a variety of ways, which included conducting joint community events incorporating educational tools, such as an inflatable replica of the inside of a colon which features interactive explanations in both English and Spanish, of various disease stages, from a normal colon to advanced CRC. The two dyads that identified deficient internal follow-up processes designed PDSAs to increase their rate of return for stool cards, with the community partner assisting with outreach to non-respondents.

### 3.4. Baseline Results from Questionnaires and Qualitative Interviews: Index of Race-Related Stress (IRRS)

Overall, 95 participants completed the IRRS, 73 from the FQHCs, and 22 from the community organizations. Only 20% of respondents from the FQHC were nL-Black. By contrast, 71% of respondents from the community organization were nL-Black. Compared to respondents from the FQHCs, proportionately more respondents from the community organization reported experiences of cultural (47% vs. 75%, *p* = 0.02) and institutional racism (30% vs. 54%, *p* = 0.04). Though not significant (*p* = 0.16), proportionately more respondents from the community organizations reported experiences of racism at the individual, compared to respondents from participating FQHCs (55% vs. 38%, respectively, Figure 1). National Survey of Primary Care Physicians’ Cancer Screening Recommendations and Practices (NSPCPCS): Though analysis is ongoing, preliminary results from 43 providers did not reveal significant provider knowledge differences by FQHC. However, there were differences identified by gender, with male providers being more likely than female providers to respond having their “payments adjusted based on their performance, as reflected in colon and cervical cancer screening reports” (*p* < 0.0001). In-Depth Qualitative Interviews: Completion and transcription of qualitative interviews is ongoing. However, 10 out of 12 qualitative interviews have been completed (Figure 1). 

### 3.5. Overall Assessment of Engagement

As of this write-up, dyads had completed a total of 10 monthly meetings, and 20 additional individual bi-weekly single dyad meetings. At least one participant from all 10 participating organizations (five FQHCs and respective community organizations) has attended every single monthly meeting and bi-weekly conversation. This assessment does not include meetings amongst dyad members outside of their regularly scheduled dyad meetings convened by VCU/MCC liaisons, or their own internal meetings where the project and its initiatives were discussed.

## 4. Discussion

Project COALESCE was designed to foster a unique dyadic partnership between FQHCs and non-clinical community organizations towards identifying and addressing structural, race-related barriers to CRC and CCa screening and follow-up. Both the selected FQHCs and respective community partners serve and represent some of the most underserved communities in southcentral VA. Some participating clinics reported baseline cancer screening rates that were barely a third of the Healthy People 2030 goal, using data just before the onset of the COVID-19 pandemic, which had a profound impact on cancer screening compliance [19]. Though not statistically different for CRC, we observed lower screening rates for Asians across both cancers. Despite the fact that only about 6% of Virginia’s population is Asian, and only 1% of the patient population in participating FQHCs identified as Asian, this finding is worth exploring, as some of the same individual and structural/structural barriers that impact other racial ethnic minorities (i.e., lack of insurance, language barriers, cultural norms, socioeconomic) may be impacting the Asian population seeking care at these FQHCs [20]. 

Though some dyads identified structural barriers, such as fractured follow-up processes impacting FIT return rates, most of the discussions remained focused on individual-level barriers (i.e., patient awareness, access, education, and fears). This focus on addressing individual-level barriers might reflect structural implicit biases, such as “organizational silence” [21], which might preclude FQHCs from identifying internal structural opportunities for change, a trend that might be pervasive in academic and clinical care. Though addressing individual-level barriers, such as transportation and copays, may be effective, others, such as increasing patient awareness and knowledge, may not be enough to sustainably increase cancer screening compliance because other structural barriers may still hinder patience compliance. For example, a disrupted clinic flow, being seen by multiple clinicians (such as in the context of an FQHC), and a lack of written materials for educational support may preclude a patient from completing their screening even after providing adequate education, and removing individual-level barriers [22]. Nonetheless, perhaps the choice to address individual-level barriers felt more within reach in the context of this newly formed dyadic partnership. Community organizations have expressed increasing their assertiveness and understanding of the value they bring to the table, as evidenced in Box 1. Therefore, given that each dyad is encouraged to go through multiple PDSA cycles, perhaps as the dyadic relationship matures, partners might be more comfortable tackling more complex structural issues in the clinic. In support of this, the continued facilitation of regular meetings and opportunities for dyads to share their findings with the larger group will be critical. Ongoing synthesis and reporting back of results from the PDSA, Fishbone, and qualitative interviews will provide further data and feedback for the dyads’ continued reflection, and the critical analysis of growth. 

Box 1Community partner’s reflection on the dyadic partner development.“At first, I felt what we were asked to do was essentially critique the health clinic, and that is ‘nice’ if it’s welcomed, but when it is someone, you don’t know, you need to have a relationship. Then I understood that what we needed to do was bring our perspective as the people who live every day in this community.”; “we started gelling”; “we began to understand their [FQHCs] priorities and they began to understand our [community organization’s] value at the table.”—Community Partner

Though academic–community partnerships have become quite common, clinical–academic–community partnerships are less common, and when done, their primary focus remains in research capacity building [23] versus improving care access and quality. Quality improvement collaboratives are also well-known; however, they generally refer to collaborations among similar organizations (i.e., between FQHCs, or partnerships with other large organizations such as the Institute for Health Care Improvement) [24]. To our knowledge, and based on partners’ feedback, this is the first time these entities have participated in a tripartite QI collaborative, especially one where the academic partner is simply a convener and facilitator, rather than the decision-maker or agenda-setter. In doing so, Project COALESCE fostered “the process of empowerment” derived from community involvement in solving community problems [25]. Also, though QI initiatives are typical within FQHCs, community members beyond those that may be represented within an FQHC board are not usually at the table for such processes, despite being the intended beneficiary of QIs. This may lead to potential blind spots or the misidentification of needs. Therefore, this dyadic process perhaps would not only bring a much more inclusive and richer perspective, but importantly, by choosing their own EBI based on their joint evaluation of the community barriers, this would lead to increased support from the community towards the implementation and long-term sustainability of the selected intervention, and the partnership. For the academic partner, this project builds on extant work in southcentral Virginia by the MCC office of Community Outreach and Engagement (COE), which complements Project COALESCE by offering free FIT cards, making navigation into the VCU Health System convenient to eligible patients, and by providing resources for screening and follow-up care for any uninsurable patient from participating clinics. These offerings from the academic institution will endure beyond the lifecycle of this project, as they are part of the broader strategic offerings integrated into MCC’s COE longitudinal plan.

### Future Directions

This unique convening of community-based clinical, non-clinical, and academic partners represents an adaptation and extension of traditional community-engaged framework. This may be a model for partnership-building among entities who share common goals, serve the same communities, have different expertise and community knowledge about existing strengths and barriers, but who have no history of partnership. The fact that participation and engagement have remained high suggests commitment to the project and its goals. The work conducted to date by the dyads has been significant, and retention has been key. Qualitative interviews will help to understand how dyads have been working as a team, the quality of the partnership developing, and which aspects have helped or hindered their progress and experience. Continued evaluation of the partnership development process will be critical for identifying best practices for building and maintaining a working relationship in the future.

## 5. Conclusions

CRC and CCa are screenable, preventable, and can be treated successfully if caught early, making it imperative that all Americans have equitable access. To identify and address the most salient structural barriers in each community, Project COALESCE has brought together clinical and non-clinical community partners. The success of bringing dyads together to complete critical QI self-analyses is encouraging. Although dyads have primarily identified EBIs that continue to address individual-level barriers, it is hoped that continued engagement in rounds of PDSA, the deepening of dyad partnerships, and facilitation support from VCU/MCC partners will provide the tools needed to step outside of their comfort zone to address structural issues that are not routinely discussed.

## Figures and Tables

**Figure 1 ijerph-19-00957-f001:**
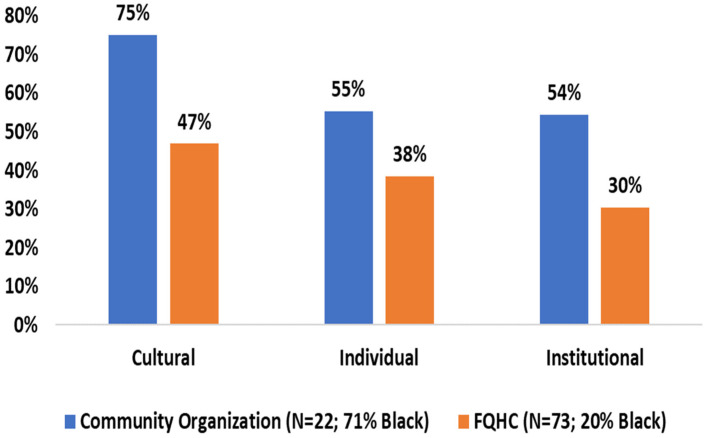
Self-reported race-related stress experiences (*N* = 95).

**Table 1 ijerph-19-00957-t001:** Comparison of average patient characteristics among participating (*n* = 5) versus non-participating FQHCs (*n* = 21), a 2-sided *t*-test *p*-value, using significance as *p* ≤ 0.05).

	Participating FQHCs	Non-Participating FQHCs	*p*-Value
	*n* = 5	*n* = 21	
Total Patients	87,419	250,801	N/A
Cervical Cancer Screening	50%	47%	0.78
Colorectal Cancer Screening	38%	40%	0.78
Children (<18 years old)	13%	24%	0.06
Adult (18–64)	74%	60%	0.01
Older Adults (age 65 and over)	13%	16%	0.44
Racial and/or Ethnic Minority	61%	41%	0.19
Hispanic/Latino Ethnicity	16%	16%	0.98
Black/African American	46%	23%	0.05
Asian	1%	2%	0.52
Best Served in another language	12%	13%	0.96
Patients at or below 100% of poverty	71%	60%	0.10
Uninsured	43%	28%	0.04
Medicaid/CHIP	18%	25%	0.19
Medicare	16%	18%	0.74

**Table 2 ijerph-19-00957-t002:** FQHC baseline data (CY2020).

	*N* = 47,707	%
Male	19,884	41.7%
Female	27,795	58.3%
White	19,528	40.9%
Black	22,268	46.7%
Latinx	5921	12.4%
Asian	456	1.0%
Other Race	3375	7.1%
Spanish-speakers	4373	9.2%
Publicly insured (Medicaid/Medicare)	25,401	53.2%
Privately insured	12,782	26.8%
Uninsured	9999	21.0%
Providers	68	0.1%
Providers who can do CCa screening	56	82.4%
Nurses	103	0.2%
Non-clinical staff	170	0.4%
Total colon cancer screening eligible	15,421	32.3%
age 50+ male	6551	32.9%
age 50+ female	8870	31.9%
age 50+ NH-White	5553	28.4%
age 50+ NH-Black	8164	36.7%
age 50+ Hispanic	1030	17.4%
age 50+ Asian	98	21.5%
age 50+ Other	576	17.1%
Total Cervical Cancer Screening Eligible	14,575	30.6%
ages 21–65 NH-White	4991	25.6%
ages 21–65 NH-Black	6766	30.4%
ages 21–65 Hispanic	2006	33.9%
ages 21–65 Asian	107	23.5%
ages 21–65 Other	705	20.9%
Total CRC screened	6337	41.1%
Male	2563	39.1%
Female	3774	42.5%
NH-White	2315	41.7%
NH-Black	3490	42.7%
Hispanic	383	37.2%
Asian	24	24.5%
Other	125	21.7%
Total CCa screened	6888	47.3%
NH-White	2431	48.7%
NH-Black	3040	44.9%
Hispanic	1087	54.2%
Asian	40	37.4%
Other	290	41.1%

## Data Availability

Data for the purposes of collaboration may be requested by contacting the first author.

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
