# Peer review of "Project COALESCE—An Example of Academic Institutions as Conveners of Community-Clinic Partnerships to Improve Cancer Screening Access"

_ijerph, 2022, doi:10.3390/ijerph19020957_

Round 1
Reviewer 1 Report
It is a very interesting study presenting crucial data regarding disparities for early cancer screening among Caucasians and non-Caucasians residing in Virginia. Studies like this are extremely needed especially due to the current pandemic has shown how significant is health inequality in the US.
Nevertheless, the article should be improved. Firstly, the article should present some data why only a small number of federal healthcare centres have decided to participate in this program, i.e., what motivated them to take part in this research project. Secondly, the clinical aspects of screening for CRC and CCa should be presented more. Thirdly, the assumption for selection bias should be made and described. In particular, some patients could primarily attend a local GP or a family doctor who is capable of organising minimally invasive screening procedures for CRC and CCa. It is not novel, that sometimes local GPs make wrong decisions due to lack of experience, absence of equipment etc. which will overall lead to misdiagnosis and cancer progression. By the time the patient reaches the federal healthcare centre, the cancer has already progressed which means that screening programs failed. Please elaborate methods how the selection bias was mitigated in this study.
Major comments
- Perhaps it will be great to see the medical data regarding methods for screening for CCa and CRC. Whether it was less invasive (cheaper) Pap smear and FOBT or more invasive (expensive) colposcopy with colonoscopy. The cost, sensitivity and specificity of abovementioned diagnostic methods is different, and it will be great to read some insights regarding which methods are preferred to be used in different racial groups.
- The percentage of Caucasians and non-Caucasian physicians working within the same clinical settings where the examined participants were studies should be elucidated.
- 5 – regarding self-reported race-related issues. The data regarding mental status and socioeconomic level of those patients should be also provided. Sometimes those factors can impact to patient’s misunderstanding, absence of rapport and as a result to race-related issues.
Minor comments
- Such words like “anecdotally” should not be used in the Journal’s Article.
Author Response
Reviewer 1:
It is a very interesting study presenting crucial data regarding disparities for early cancer screening among
Caucasians and non-Caucasians residing in Virginia. Studies like this are extremely needed especially due
to the current pandemic has shown how significant is health inequality in the US.
Author response: Thank you!
Nevertheless, the article should be improved. Firstly, the article should present some data why only a small
number of federal healthcare centres have decided to participate in this program, i.e., what motivated them
to take part in this research project.
Author response: The small number of participating FQHCs reflects the pilot nature of the project, rather
than non-participation (opt-out) of facilities. Only five FQHCs were recruited. As stated in the manuscript,
the 5 FQHCs were chosen based on “2018 HRSA data to identify five FQHCs in Virginia with CRC and CCa
screening rates well below the state’s average (70% and 84% respectively) and the Healthy People 2030
goals for these cancers (74% and 84% respectively). We ensured these FQHCs minimally represented or
surpassed the racial and ethnic (19% Black, 10% Latinx) and geographic (25% rural) diversity of the state”
Secondly, the clinical aspects of screening for CRC and CCa should be presented more.
Author response: We have added the following sentence to the bottom of the first paragraph in the
introduction section (line 48) to clarify the types of screenings done for each of these cancers: “Yet,
prevention and early detection of these cancers is possible through screenings that detect and/or remove
pre-cancerous lesions (Papanicolaou for CCa and stool-based (i.e., FIT or FOBT) or visual (structural)
exams (i.e., colonoscopy, flexible sigmoidoscopy) for CRC).
Thirdly, the assumption for selection bias should be made and described. In particular, some patients could
primarily attend a local GP or a family doctor who is capable of organising minimally invasive screening
procedures for CRC and CCa. It is not novel, that sometimes local GPs make wrong decisions due to lack of
experience, absence of equipment etc. which will overall lead to misdiagnosis and cancer progression. By
the time the patient reaches the federal healthcare centre, the cancer has already progressed which means
that screening programs failed. Please elaborate methods how the selection bias was mitigated in this
study.
Author response: Federally Qualified Health Centers in the United States provide primary care to patients
in the community regardless of their ability to pay. They are not referral sites, as suggested by the
reviewer’s comment. Their federal designation (provided by the Health Resources and Services
Administration, HRSA) means they receive additional federal dollars to support care provision regardless
of insurance status or finances. Therefore, FQHCs are more likely to serve publicly insured, uninsured,
underinsured or uninsurable (i.e., non-US citizens) individuals. We have added a sentence to the end of the
last paragraph in the introduction clarifying what is an FQHC: “FQHCs are community-based clinics that
receive funds from the Health Resources and Services Administration (HRSA) to provide primary care
regardless of a patient’s ability to pay.”
Reviewer 1 Major comments:
Perhaps it will be great to see the medical data regarding methods for screening for CCa and CRC. Whether
it was less invasive (cheaper) Pap smear and FOBT or more invasive (expensive) colposcopy with
colonoscopy. The cost, sensitivity and specificity of abovementioned diagnostic methods is different, and it
will be great to read some insights regarding which methods are preferred to be used in different racial
groups.
Author response: While the cost, sensitivity, specificity and differential uptake by racial group are beyond
the scope of this process manuscript, we added under “FQHC Baseline Data – Updated for Participating
Clinics” some verbiage to address the type of testing available at the participating FQHCs, as well as
proportion of providers able to perform these screenings on-site as follows: “They reported employing 170
non-clinical staff, 103 nurses, and 68 providers, a majority (82% of providers) of which performed on-site
Pap screening for CCa, all provided stool cards (predominantly Fecal Immunochemical Test, FIT cards) for
CRC.
The percentage of Caucasians and non-Caucasian physicians working within the same clinical settings
where the examined participants were studies should be elucidated.
Author response: We added a sentence to report provider race/ethnicity data for the 2 clinics that
reported such data as follows: “FQHC Baseline Data – Updated for Participating Clinics”: “Of two clinics who
reported the racial/ethnic composition of their providers, the clinics with the second and third largest
proportion of nL-Black patient population (62% and 44% nL-Blacks) reported that 0% and 13% of their
providers respectively identified as nL-Blacks.”
The data regarding mental status and socioeconomic level of those patients should be also provided.
Sometimes those factors can impact to patient’s misunderstanding, absence of rapport and as a result to
race-related issues.
Author response: While we agree with the reviewer that these individual-level factors may correlate with
screening uptake, such information is beyond the scope of this manuscript, which is intended to describe
the development process for the initiative. Additionally, mental health status might not be routinely
collected for all patients, as well as socioeconomic level, beyond the proxies of insurance status, education,
and perhaps the patient’s address.
Minor comments: Such words like “anecdotally” should not be used in the Journal’s Article.
Author response: The word “anecdotally” was used to refer to anecdotal evidence, which is qualitative
data. However, we appreciate the reviewer’s feedback and have removed the word, as we had included the
stated narrative in Box 1.
Reviewer 2 Report
Dear authors, please find below some suggestions:
2.7. Data analysis
Line 218: authors report methods to summarize and analyze quantitative data but in results section there are not these information (Student t-test and what as descriptive index? Mean and sd or other descriptive statistics?).
Pearson Chi square test is used for categorical variables; with proportional variables the authors mean proportion? Is alpha at 0.05 the statistical significance threshold? Which software was used to perform data analysis?
- Results
Line 227: authors stated that FQHCs were “more likely”.. if these results refer to Chi-square test is not appropriate to verify that one condition is more likely but it would be correct to describe data, this result shows that in FQHCs group the proportion of people under 65 years old was slightly higher compare to non-participating FQHCs (74% vs 60%, p=0.01).
Line 233: why is there a bibliographic citation in results section?
3.5 section: the description of the results is not clear, especially about figure 1, authors state that the comparison is between FQHCs respondents and community organization but in x-axis of the graph there are stress conditions. As stated above, terms “more likely” is not appropriate for this type of analysis and test, these results show only the distribution of two variables (from a bivariate frequency distribution table).
Author Response
Reviewer 2:
Line 218: authors report methods to summarize and analyze quantitative data but in results section
there are not these information (Student t-test and what as descriptive index? Mean and sd or other
descriptive statistics?). Pearson Chi square test is used for categorical variables; with proportional
variables the authors mean proportion? Is alpha at 0.05 the statistical significance threshold?
Which software was used to perform data analysis?
Author response: We have clarified under the methods section of this process paper that “As data
collection is ongoing, herein we provide the initial, univariate analysis of baseline data for year one
of the project.” We also added the use of StataIC 13 software for quantitative analysis.
Line 227: authors stated that FQHCs were “more likely”.. if these results refer to Chi-square test is
not appropriate to verify that one condition is more likely but it would be correct to describe data,
this result shows that in FQHCs group the proportion of people under 65 years old was slightly
higher compare to non-participating FQHCs (74% vs 60%, p=0.01).
Author response: We thank the reviewer for this statement. We have corrected to state “served a
higher proportion of patients that were…”
Line 233: why is there a bibliographic citation in results section?
Author response: The citation refers the reader to the source for the stated Healthy People 2030
goals.
3.5 section: the description of the results is not clear, especially about figure 1, authors state that
the comparison is between FQHCs respondents and community organization but in x-axis of the
graph there are stress conditions. As stated above, terms “more likely” is not appropriate for this
type of analysis and test, these results show only the distribution of two variables (from a bivariate
frequency distribution table).
Author response: We thank the reviewer for their comments. For figure one, the comparison is
indeed between the FQHCs and the community organizations, represented by bars in two colors
(orange and blue, respectively). The graph reports these three bars across three domains
measured: cultural, individual, and institutional. While we could substitute the graph for the one
below, the authors believe it would be simpler for the readers to keep the results grouped by
domain on the X axis, rather than by type of organization (community organization versus FQHC).

Reviewer 3 Report
This is an interesting contribution addressing a relevant issue on healthcare access equality accross a same country. Similar concers are observed worlwide among different communities within a single country, e.g, elderly, immigrants, or people living in large innercities vs those living in suburban areas or isolated in mountains or field areas.
With regards to that, the COALESCE projet looks suitable for publication. It will retain attention of many readers.
The submission is however not very easy to read, since the are no tables and only one figure. The manuscript may be substantially improved while adding a few tables to highlights the result section
Author Response
Reviewer 3:
This is an interesting contribution addressing a relevant issue on healthcare access equality accross a same
country. Similar concers are observed worlwide among different communities within a single country, e.g,
elderly, immigrants, or people living in large innercities vs those living in suburban areas or isolated in
mountains or field areas.
Author response: Tbank you!
With regards to that, the COALESCE projet looks suitable for publication. It will retain attention of many
readers. The submission is however not very easy to read, since the are no tables and only one figure. The
manuscript may be substantially improved while adding a few tables to highlights the result section
Author response: We have added an additional table (Table 1) to better describe the HRSA data “FQHC
Baseline Data – HRSA, 2019”
Round 2
Reviewer 1 Report
I believe that authors have responded to all my previous comments and the manuscript is ready for publication.
Author Response
We kindly thank the reviewer for taking the time to review our manuscript. Your comments have certainly made our manuscript better.
Reviewer 2 Report
Dears authors, thanks for the changes you have made in the manuscript.
Some minor suggestions:
- Line 219: please specify that alpha at 0.05 is the statistical significance threshold
- Figure 1 shows distribution of community and FQHC stratifying by Cultural, individual and institutional subgroups. Is it correct that the two bars do not reach 100%? Lines 349,350 : the terms “more likely” are not appropriate for this type of test, these results show only the distribution of two variables (please modify it).
Author Response
We kindly thank the reviewer for taking the time to review our manuscript once again. Your comments continue to improve our manuscript.
- Line 219: please specify that alpha at 0.05 is the statistical significance threshold - RESPONSE: I have added verbiage to clarify that alpha 0.05 is the statistical significance threshold.
- Figure 1 shows distribution of community and FQHC stratifying by Cultural, individual and institutional subgroups. Is it correct that the two bars do not reach 100%? RESPONSE: No. The figure only depicts those from either the community organization or the FQHC that reported experiencing either cultural, individual or institutional race-related stress. Therefore, the bars shown should not total 100%. For example, for experiences of cultural race-related stress from the community organization (N=22), 75% reported having such experiences, thus 25% (not shown) did NOT report having those experiences. For simplicity, only the affirmative response is shown. We hope this clarifies for the reviewer.
- Lines 349,350 : the terms “more likely” are not appropriate for this type of test, these results show only the distribution of two variables (please modify it). RESPONSE: The language has been changed to specify that "proportionately more respondents from... reported xyz". We hope this change is satisfactory to the reviewer.